# OpenReview forum: "MECAT: A Multi-Experts Constructed  Benchmark for Fine-Grained Audio Understanding Tasks"
_ICLR.cc/2026/Conference — Submitted to ICLR 2026_

### Official Review · Reviewer_jZBM · 2025-10-31

**Soundness:** 2
**Presentation:** 2
**Contribution:** 2
**Rating:** 2
**Confidence:** 4

**Summary:**

This submission proposes a novel benchmark, MECAT, for finegrained open-ended audio understanding evaluation. MECAT is constructed with a complex pipeline with many AI experts in the loop, including annotation, aggregation, filtering, and more. The paper claims MECAT is more diverse than previous benchmarks and has higher quality. In addition, the paper proposes a new method for open-ended captioning evaluation.

**Strengths:**

The proposed benchmark is very useful to the community as it reveals many detailed aspects of the quality of audio language models. Especially, there are sub-metrics with focus on different domains, lengths and cognitive categories, which could shed light on current audio language models and guides the community to develop better models accordingly. The AI expert annotations are also well-designed such that the community can easily reproduce such samples with an automated pipeline.

**Weaknesses:**

For the data construction, since all steps are based on AI models and there is no human-in-the-loop, it is questionable whether the outputs are high-quality enough or have some hidden bias. Since none of the AI models are super reliable to my knowledge, I tend to believe such biases or inaccurate test samples exist. Therefore, this work should at least verify the quality of the testset via human inspection or tests. Furthermore, there is no solid evidence the proposed benchmark is siginificantly more useful than MMAU/MMAR, despite it being open-ended.

For the metric design part, while I agree that current captioning-based metrics (CIDEr, SPICE, FENSE, etc) are not good enough, I do not think the proposed DATE addresses the issues. First, the TF-IDF based $v_T$ computation does not capture the temporal information because it is a sum $\sum_{t\in T}$. It is therefore more similar to a bag-of-word sementic embedding, which is undesirable in captioning evaluation. The Setence-BERT embedding also does not consider the complex meaning of particular words given the context, as the specific use of some word can be very different from its "averaged" meaning as represented by $E(\cdot)$. Moreover, the cross-sample discriminability considers the whole audio testset pool, making the metric not per-sample static and dependent on the entire testset. However, we expect the metric to be fixed for a given sample, regardless what the other samples are.

For the experimental design part, only very few baselines are evaluated. This is not acceptable for a benchmark-focused paper -- which should check all existing open models as a bottom line (see how many baselines MMAR reported). There is not much analysis on different audio language models given the scores. What do the scores imply besides simply ranking the models? Do the scores reveal any potential bias of existing models? Do the scores show noticeable hints that prior benchmarks (MMAU, MMAR) did not show? Are there PEFT or training-free methods to improve baseline models on the benchmark? These are some examples that can show the usefulness of the proposed benchmark.

**Questions:**

Please refer to the weakness section.

---

> ### Author Response · Authors · 2025-11-29
> **Point-by-point response to Reviewer jZBM -- Part1**
>
> **Response**: We thank the reviewer for the valuable comments.
>
> > Reviewer jZBM (Weakness 1): For the data construction, since all steps are based on AI models and there is no human-in-the-loop, it is questionable whether the outputs are high-quality enough or have some hidden bias. Since none of the AI models are super reliable to my knowledge, I tend to believe such biases or inaccurate test samples exist. Therefore, this work should at least verify the quality of the testset via human inspection or tests. Furthermore, there is no solid evidence the proposed benchmark is siginificantly more useful than MMAU/MMAR, despite it being open-ended.
>
> Please refer to the Response to **the shared concern (ii) Data quality & potential bias** and Response to the **shared concern (iv) Scope and positioning of MECAT**. Briefly, on the construction side, MECAT labels are produced by a multi-expert → CoT pipeline that explicitly detects and filters conflicts across diverse expert models, and a 150-pair human A/B study (instruction “accuracy first, then level of detail”) shows that MECAT references are strongly preferred over generic/incorrect alternatives and are on par with human-written captions.
>
> Regarding the compared with MMAU/MMAR, MMAU/MMAR are primarily **multiple-choice QA benchmarks**, where models can exploit answer options (and even non-semantic cues) without generating grounded descriptions. MECAT instead uses **open-ended captioning and QA with dense, multi-perspective references**, and evaluates them with DATE, which jointly rewards semantic accuracy and cross-sample distinctiveness. This open-ended, discriminative setup makes it harder for models to succeed with generic templates or option-guessing and surfaces failure modes (e.g., template reuse, hallucinations) that are difficult to detect under multiple-choice evaluation.
>
> > Reviewer jZBM (Weakness 2) For the metric design part, while I agree that current captioning-based metrics (CIDEr, SPICE, FENSE, etc) are not good enough, I do not think the proposed DATE addresses the issues. First, the TF-IDF based $v_T$ computation does not capture the temporal information because it is a sum $\sum_{t \in T}$. It is therefore more similar to a bag-of-word sementic embedding, which is undesirable in captioning evaluation. The Setence-BERT embedding also does not consider the complex meaning of particular words given the context, as the specific use of some word can be very different from its "averaged" meaning as represented by $\mathbf{E}$ . Moreover, the cross-sample discriminability considers the whole audio testset pool, making the metric not per-sample static and dependent on the entire testset. However, we expect the metric to be fixed for a given sample, regardless what the other samples are.
>
> Please refer to the Response to the **shared concern (iii) DATE design and alignment with human judgments** for a detailed rationale of the $\text{tf–idf}$ weighting, contextual Sentence-BERT embeddings, and the corpus-relative discriminability term. Briefly, DATE is designed to address two concrete failure modes we observe in MECAT: generic-but-vague captions that are reused across many clips and over-specific but incorrect captions. To do this, DATE combines a semantic similarity term with a cross-sample discriminability term via the harmonic mean, so captions must be both reference-faithful and clip-specific to obtain a high score.
>
> Regarding the concern that our $\text{tf–idf}$ computation acts like a Bag-of-Words (BoW) model, the revised paper now clarifies (and adds Appendix E) that $v_T$ is built using embedding-level TF–IDF on contextual Sentence-BERT token embeddings: each token embedding $E(t)$ is produced by a BERT-like encoder and already encodes its sentence context, so $v_T = \sum_{t\in T} w(t,T)E(t)$ is not a simple bag-of-words over static word-type vectors. To empirically confirm this, we performed an ablation: shuffling the words of a caption significantly decreased the DATE score (Original $\text{DATE}=62.3$ vs. Shuffled $\text{DATE}=35.0$), definitively proving that **the metric is sensitive to word order and sequential structure**.
>
> Regarding the limitations of Sentence-BERT, We agree that any embedding-similarity-based term has limitations (e.g., smoothing over some fine-grained semantic or temporal distinctions), which we explicitly acknowledge. This is precisely why DATE adds the discriminability component rather than relying on similarity alone.
>
> Regarding “per-sample staticity”, DATE is intentionally corpus-relative: detecting generic template reuse necessarily requires comparing a caption against other clips in the benchmark—a feature we view as essential for measuring distinctiveness rather than as a defect. For a fixed benchmark release and model outputs, the corpus is fixed, making all per-sample DATE scores deterministic.

---

> ### Author Response · Authors · 2025-11-29
> **Point-by-point response to Reviewer jZBM -- Part2**
>
> > Reviewer jZBM (Weakness 3) For the experimental design part, only very few baselines are evaluated. This is not acceptable for a benchmark-focused paper -- which should check all existing open models as a bottom line (see how many baselines MMAR reported). There is not much analysis on different audio language models given the scores. What do the scores imply besides simply ranking the models? Do the scores reveal any potential bias of existing models? Do the scores show noticeable hints that prior benchmarks (MMAU, MMAR) did not show? Are there PEFT or training-free methods to improve baseline models on the benchmark? These are some examples that can show the usefulness of the proposed benchmark.
>
> Please see Response to the **shared concern (i) Insufficient baselines** for details on the expanded model set (15 systems, including Audio Flamingo 3, Gemini 2.5 Pro/Flash, Qwen-Omni, and Phi-4-Multimodal) and the new slice-wise diagnostics (domain; Pure/Mixed; Short/Long; QA categories).
>
> Regarding the analysis of the extended model sets, In the revised paper, **Section 6.3.1 (see line 439-461)** adds an overall analysis of what the scores imply: on MECAT-Caption, LALMs markedly outperform caption-only baselines, Music/Sound performance drops by roughly 10–25\% from pure to mixed domains, revealing a speech-centric bias, and all models remain weak on Content-Unrelated captions, indicating that acoustic quality and environment are still poorly modeled. On MECAT-QA, the hierarchy largely mirrors captioning but the gap between proprietary and open-weight models narrows, and we observe a clear capability dichotomy: strong on Direct Perception and content-based Inference, but consistently low on Quality Assessment and Environment Reasoning.
>
> **Section 6.3.2 (see lines 462–518)** then drills down on two concrete behaviors. On the Pure Speech subset, DATE cleanly separates models whose captions are generic but high-similarity from those that produce clip-specific, discriminative descriptions: Gemini’s lead comes from combining very high similarity with the strongest discriminability, while Qwen3-Omni reaches a comparable rank mainly via discriminability despite more moderate similarity. On silent segments, many LALMs hallucinate fluent but spurious speech (e.g., `I’m gonna be a daddy`, `Thank you`), whereas only a few models reliably abstain when acoustic evidence is absent. These analyses show that DATE does more than rank systems: it surfaces generic-template behavior, speech bias, and silence-hallucination failure modes that multiple-choice benchmarks such as MMAU/MMAR tend to under-emphasize; please also see Response to the **shared concern (ii) Data quality & potential bias** for how MECAT complements MMAU/MMAR.
>
> Regarding directions for improving current LALMs, we see the main opportunity in the supervision signal itself. Our results indicate that existing systems are often evaluated primarily on safe, high-level summaries, and are much less frequently encouraged to produce distinct, clip-specific captions instead of generic templates, and describe content-unrelated acoustic properties. MECAT (together with DATE) is therefore intended to serve as a diagnostic benchmark and a source of richer supervision that promotes such fine-grained, acoustically grounded behaviors, while model-specific optimization strategies are complementary and lie outside the scope of this work.

---

### Official Review · Reviewer_sMGA · 2025-11-01

**Soundness:** 2
**Presentation:** 2
**Contribution:** 2
**Rating:** 4
**Confidence:** 5

**Summary:**

The paper proposes MECAT, a Multi-Expert Constructed Benchmark for Fine-Grained Audio Understanding Tasks. The benchmark comes with a novel evaluation metric: DATE (Discriminative-Enhanced Audio Text Evaluation). The metric penalizes generic terms and rewards detailed descriptions by combining single-sample semantic similarity with cross-sample discriminability. Audios are collected from ACAV100M and the benchmark consists of 20,000 Creative Commons-licensed audio clips, each with a maximum duration of 10 seconds.

**Strengths:**

- I appreciate the use of open models for benchmark curation.
- The results and discussion section is nice and appreciated. Provides useful insights.
- DATE is novel and well motivated.

**Weaknesses:**

- The maximum duration is only 10 seconds. Short audio snippets can only be as complex.
- The comprehensive of models evaluated is not great. Many more models have been released, including Audio Flamingo 3, etc
- The tasks for coverage are very foundational and have been explored earlier. So I find a slight lack of novelty here.

**Questions:**

- Why were general sounds not included? Any specific reason?
- What is the impact of the CoT reasoning?

---

> ### Author Response · Authors · 2025-11-29
> **Point-by-point response to Reviewer sMGA**
>
> **Response**: We thank the reviewer for the valuable comments.
>
> > Reviewer sMGA (Weakness 1): The maximum duration is only 10 seconds.
>
> Please see Response to the **shared concern (iv) Scope and positioning of MECAT (10-second clips and relation to MMAU/MMAR)**. Briefly, MECAT is deliberately designed as a dense, clip-local benchmark: each 10-second clip carries 18 multi-perspective captions and 5 open-ended QA pairs, which fits current LALM context limits, isolates salient events without long-range confounds, and makes clip-level cross-sample discriminability well-defined for DATE. Long-audio leaderboards primarily test robustness over extended sequences; MECAT complements them by stressing richness and specificity of local understanding.
>
> > Reviewer sMGA (Weakness 2): Comprehensiveness of models evaluated is not great.
>
> Please see Response to the **shared concern (i) Insufficient baselines**. In the revised experiments, we expand to 14+ systems, including proprietary models (Gemini 2.5 Pro/Flash) and many open LALMs (Qwen2.5-Omni 3B/7B, Qwen3-Omni, Audio Flamingo 3, Kimi-Audio-7B, Baichuan-Audio/Omni, Phi-4-Multimodal, step-audio-2-mini, Mimo-Audio), as well as caption-only baselines (EnClap, Pengi), and update the caption/QA leaderboards accordingly.
>
> > Reviewer sMGA (Weakness 3): Tasks are foundational; slight lack of novelty.
>
> We thank the reviewer for this comment. While we agree that the underlying tasks (audio captioning and QA) are foundational, we would like to emphasize that the **novelty of our work lies in how these tasks are annotated and evaluated**, rather than in defining new task types. As discussed in Section 1 **(see line 43-64 of the revised paper)**, current audio benchmarks suffer from an `evaluation gap`: on the data side, annotations are often event-level, single-perspective, and repeatedly reused across AudioSet-derived benchmarks, and on the metric side, standard lexical and embedding-based scores tend to reward generic captions (e.g., “A dog is barking and people are talking”) even for semantically distinct scenarios. This makes it difficult to distinguish models with true perceptual accuracy from those producing vague outputs.
>
> MECAT addresses this evaluation gap by (i) **providing fine-grained, multi-perspective, domain-stratified annotations** via the multi-expert and CoT synthesis pipeline, and (ii) introducing **DATE, a discriminative, corpus-relative metric** that jointly evaluates semantic accuracy and cross-sample distinctiveness. Together, these components expose generic-boilerplate and over-specific-hallucination failure modes that are difficult to detect in prior single-reference or multiple-choice benchmarks such as MMAU/MMAR.
>
> > Reviewer sMGA (Question 1): Why were general sounds not included?
>
> General sounds are included in MECAT. As described in Section 3.1 **(see line 191-198 of the revised paper)** and Figure 2 **(see line 200-214 of the revised paper), MECAT covers both pure Sound and mixed cases (i.e., Speech+Sound, Music+Sound, Speech+Music+Sound), with events such as alarms, machinery, animal calls, and transport/impact sounds.
>
> > Reviewer sMGA (Question 2): What is the impact of the CoT reasoning?
>
> Please see Response to the **shared concern (ii) Data quality & potential bias**. Briefly, the CoT synthesis stage takes the outputs of ASR, diarization, and event/music taggers, explicitly detects conflicts, and either reconciles them or drops unsupported claims.

---

### Official Review · Reviewer_5m6Q · 2025-11-04

**Soundness:** 2
**Presentation:** 3
**Contribution:** 2
**Rating:** 6
**Confidence:** 5

**Summary:**

The paper introduces MECAT, a benchmark for fine-grained audio understanding with two tasks, captioning and open-set QA. Data are built by a multi-expert pipeline that runs domain-specific audio models, then uses an LLM with chain-of-thought to synthesize rich captions and QA pairs. The benchmark spans pure and mixed domains across speech, music, sound events, silence, and their combinations. The paper also proposes DATE, a metric that blends weighted semantic similarity and cross-sample discriminability to reward specific, distinctive descriptions.

**Strengths:**

1. The paper is easy to read. Task definitions, pipeline diagrams, and scoring formulas are explicit. Caption scoring lists categories and weights, and QA aggregation is uniformly specified. This clarity lowers the barrier for reimplementation and for fair comparison.

2. Specialized audio models first detect speech content, speakers, musical attributes, events, and acoustic conditions. A language model then synthesizes these signals into captions and QA with step by step reasoning. This staged design helps surface details that single labelers often miss and supports mixed clips that blend speech, music, and environmental sounds.

3. DATE combines a tf-idf weighted semantic similarity term with a cross-sample discriminability term, joined by a harmonic mean. This discourages generic captions and favors precise descriptions that fit the target clip better than others.

**Weaknesses:**

1. The construction stack depends on ASR, diarization, emotion, and event taggers. Mistakes at these stages can leak into captions and QA despite later filtering. The paper notes filtering, yet residual noise is likely in complex audio. This critiques robustness, not the clarity of the written specification.

2. Audio Flamingo 2 appears in analysis during construction and also in evaluation. This can advantage that model family. A clearer separation between builder models and evaluated systems would reduce this risk.

3. Caption weights are fixed, and the QC pipeline uses an empirical GLAP threshold. The formulas are clear, yet the motivation for these constants is brief, which can affect reproducibility across domains.

**Questions:**

1. What design principle determined the caption category and subcategory weights, and are these weights intended to reflect perceived task importance or observed reliability of references. Please clarify whether weights are constant across domains and why.

2. Please explain why DATE combines a tf–idf weighted semantic similarity term with a cross-sample discriminability term, and why you fuse them with a harmonic mean. Which failure modes in audio captioning does this design aim to fix, and why prefer these components over alternatives such as pure embedding similarity or learned weighting.

---

> ### Author Response · Authors · 2025-11-29
> **Point-by-point response to Reviewer 5m6Q**
>
> **Response**: We thank the reviewer for the valuable comments.
>
> > Reviewer 5m6Q (Weakness 1): The construction stack depends on ASR, diarization, emotion, and event taggers. Mistakes at these stages can leak into captions and QA despite later filtering. The paper notes filtering, yet residual noise is likely in complex audio. This critiques robustness, not the clarity of the written specification.
>
> Please see Response to the **shared concern (ii) Data quality & potential bias**. Briefly, MECAT labels are produced by a multi-expert + CoT synthesis pipeline that explicitly detects conflicts between expert signals and either reconciles or discards unsupported claims; in a quantitative audit, fewer than 0.5\% of released clips showed any potential unresolved conflict, and a 150-pair human A/B study further confirmed that MECAT references are strongly preferred over generic/incorrect alternatives and are on par with human-written captions under an “accuracy first, then level of detail” instruction.
>
> > Reviewer 5m6Q (Weakness 2): Audio Flamingo 2 appears in analysis during construction and also in evaluation. This can advantage that model family. A clearer separation between builder models and evaluated systems would reduce this risk.
>
> We agree this could suggest circularity. In the revised paper, we include Audio Flamingo 3 as the main Audio Flamingo-family baseline and keep Audio Flamingo 2 only as a clearly flagged reference row. The tables explicitly mark that Audio Flamingo 2 was used as one expert signal in the construction pipeline, and that Audio Flamingo 3 comes from the same family, so Audio Flamingo-family numbers are not misinterpreted as fully independent of the construction stack. **(See line 468-485 for Table 2 and line 486-502 for Table 3 of the revised paper)**
>
> > Reviewer 5m6Q (Weakness 3): Caption weights are fixed and GLAP is empirical; motivation is brief.
>
> > Reviewer 5m6Q (Question 1): What design principle determined the weights? Are they constant across domains?
>
> Please see Response to the **shared concern (v) Task-level weighting**. Briefly, all subtask scores are reported before any weighting, and the aggregated score uses fixed coefficients chosen to reflect task salience and data distribution. The GLAP threshold is empirical by design and is chosen conservatively so that dubious pairs are filtered out rather than retained.
>
> > Reviewer 5m6Q (Question 2): Why DATE = H(tf-idf–weighted similarity, cross-sample discriminability)? What failure modes does this fix vs. pure embedding similarity or learned weighting?
>
> Please see Response to the **shared concern (iii) DATE design and alignment with human judgments**. Briefly, DATE uses a tf–idf–weighted semantic similarity term to measure how well a caption matches its own clip’s references, and a cross-sample discriminability term to penalize captions that would also fit many other clips; the harmonic mean forces both semantic accuracy and clip-specific distinctiveness to be high, directly addressing generic-but-vague and over-specific-but-wrong failure modes that pure embedding similarity or learned weighting may fail to capture.

---

### Official Review · Reviewer_n5Pw · 2025-11-05

**Soundness:** 3
**Presentation:** 2
**Contribution:** 2
**Rating:** 4
**Confidence:** 4

**Summary:**

Compared to human beings, audio language models fall short of human-level audio understanding. The author points out that one of the reasons for this is that current benchmarks developed for audio LMs have significant limitations on annotations and metrics. They constructed a benchmark, called MECAT, and they also proposed a new metric. Then, the author evaluated SOTA audio LMs on this new benchmark to demonstrate their capabilities and limitations.

**Strengths:**

The introduction and explanation of machine hearing and the limitations of the current metric are clear.

The less fine-grained labeling problems existing in current captioning and question-answer benchmarks are explained in detail.

I also like the idea of the distinct audio domains designed. The domain experts are explained in detail.

**Weaknesses:**

**1.** The structure of some content can be improved. At the end of line 44, the authors state that the benchmark is often a neglected bottleneck. But right after it, they started to discuss the limitations of the current metric. I would suggest putting the following paragraph, the one started on line 53, right after it to ensure consistency and better presentation.

**2.** Figure 1 has too many written descriptions. I would suggest a sub-figure that contains all the explanations for clarity.

**3.** For audio understanding tasks, one of the problems in current audio LMs is that the length of audio they can process (before a significant performance drop) is very short, usually less than 30 seconds. Since this work focuses on understanding more nuanced, more descriptive content for each audio clip, I think the 10-second audio length in MECAT is very limited. This is one major drawback from the perspective of the contribution of the work.

**4**. The discussion on the limitations of the work is too superficial and shows limited insight.

**Questions:**

**1.** In Appendix D, the multiple choice options under section 3.1.1, how are these selected?

**2.** So, the overall objective of 'synthesize' is for labeling or some other purposes?

**3.** What will DATE represent, or mean, on a dataset level?

---

> ### Author Response · Authors · 2025-11-29
> **Point-by-point response to Reviewer n5Pw -- Part1**
>
> **Response**: We thank the reviewer for the valuable comments.
>
> > Reviewer n5Pw (Weakness 1): The structure of some content can be improved. At the end of line 44, the authors state that the benchmark is often a neglected bottleneck. But right after it, they started to discuss the limitations of the current metric. I would suggest putting the following paragraph, the one started on line 53, right after it to ensure consistency and better presentation.
>
> We thank the reviewer for carefully catching this structural issue. In the submitted version, an intermediate paragraph on annotation limitations was accidentally dropped during editing, which indeed made the transition from `benchmark as a bottleneck` directly into `annotation and metric limitations` abrupt.
>
> **Section 1 (see line 47-53 of the revised paper)** is corrected to first discuss data-annotation limitations and then metric limitations, **following this logit: `benchmark bottleneck` → `annotation issues` → `metric issues`**. Concretely, the relevant passage now reads:
>
> ` ... a crucial and often-overlooked bottleneck is the existing evaluation benchmark.`
>
> `The first challenge lies with data annotations, ...`
>
> `The second challenge is rooted in evaluation metrics. ...`
>
> We hope this improves the readability of the introduction.
>
> > Reviewer n5Pw (Weakness 2): Figure 1 has too many written descriptions. I would suggest a sub-figure that contains all the explanations for clarity.
>
> **Figure 1 (see line 135-161 of the revised paper)** is split into two vertically stacked panels: (A) an overview of the proposed annotation construction pipeline, and (B) a sub-figure with concise task explanations and representative examples for audio captioning and QA.
>
> > Reviewer n5Pw (Weakness 3): 10-second audio length is very limited for nuanced understanding.
>
> Please refer to the Response to the **shared concern (iv) “Scope and positioning of MECAT (10-second clips and relation to MMAU/MMAR)”**. Briefly, MECAT is intentionally designed as a dense, clip-local benchmark: each 10-second clip carries 18 multi-perspective captions and 5 open-ended QA pairs, which target fine-grained understanding within a short context. The 10-second length fits current LALM context limits, isolates salient events without long-range confounds, and makes cross-sample discriminability at the clip level well defined for DATE.
>
> > Reviewer n5Pw (Weakness 4): Limitations are too superficial.
>
> We agree and have substantially expanded the Limitations section in the revised manuscript. We now explicitly (i) state the use of short (≤10 s) clips as a core constraint and note that MECAT is complementary to long-audio/streaming benchmarks, (ii) discuss that DATE, while improving over standard metrics, still inherits fundamental limitations of embedding-based similarity (e.g., potentially high scores when key entities are swapped), and (iii) point out that DATE is most informative when the test set contains diverse audio clips and is less suitable for highly homogeneous audio. **The revised Limitations in Section 7 reflects this expanded discussion and outlines future work on longer contexts and more fine-grained semantic evaluation (see line 535-539 of the revised paper)**.

---

> > ### Author Response · Authors · 2025-11-29
> > **Point-by-point response to Reviewer n5Pw -- Part2**
> >
> > > Reviewer n5Pw (Question 1): In Appendix D, the multiple-choice options under section 3.1.1—how are these selected?
> >
> > In Appendix D, Section 3.1.1 **(see line 958-969 of the revised paper)**, the items listed under `Identify dominant characteristics of this audio` are not evaluation-time multiple-choice options, but examples types of salient content used in an internal CoT prompting step. At this stage, the LLM is asked to decide, based on the audio and expert analyses, which aspects are most salient for this clip (e.g., the main spoken topic, a sudden sound event, a dominant musical motif, an unusual acoustic scene, or notable quality issues), and to summarize these dominant characteristics to guide subsequent caption and QA generation.
> >
> > > Reviewer n5Pw (Question 2): Is the objective of `synthesize` for labeling or other purposes?
> >
> > The objective of `synthesize` is labeling. As illustrated in Figure 1 (A) **(see line 135-149 of the revised paper)** and Section 4.2 **(see 292-299 of the revised paper)**, the CoT synthesis stage takes the expert signals (ASR, tags, diarization, music descriptors, acoustic analyzers, etc.), reconciles conflicts, and outputs the final reference captions and open-ended QA (with confidence tags). Only these synthesized labels are used at evaluation time; the intermediate synthesis reasoning itself is not used for any downstream purpose.
> >
> > > Reviewer n5Pw (Question 3): What does DATE represent at the dataset level?
> >
> > Please refer to the Response to the **shared concern (iii)** DATE design and alignment with human judgments. Briefly, at the sample level DATE is the harmonic mean of a tf–idf–weighted semantic similarity term (caption accuracy w.r.t. its own references) and a cross-sample discriminability term (how specific the caption is to that clip within MECAT). Dataset-level DATE is simply the average of per-sample DATE over all clips, so a higher dataset-level DATE indicates that a system, on average, produces captions that are both accurate with respect to MECAT references and non-generic across the corpus.

---

### Official Review · Reviewer_iKJk · 2025-11-05

**Soundness:** 2
**Presentation:** 3
**Contribution:** 2
**Rating:** 2
**Confidence:** 4

**Summary:**

The paper introduces MECAT, a Multi-Expert Constructed Benchmark for Fine-Grained Audio Understanding Tasks. It is generated through a pipeline that integrates analysis from specialized expert models with COT large language model reasoning. It provides multi-perspective,
fine-grained captions and open-set question-answering pairs. The paper also introduces a novel metric: DATE (Discriminative-Enhanced Audio Text Evaluation) which penalizes generic terms and rewards detailed descriptions by combining single-sample semantic similarity with cross-sample discriminability.

**Strengths:**

- The paper introduced tf-idf–weighted embeddings + cross-sample rank to penalize generic captions
- Novel use of specialized audio-related experts models, including content-specific models and content unrelated models followed by Chain-of-Thought enhanced LLM reasoning for caption generation
- MECAT provides 18 reference captions per clip and reports a vocabulary of 22,595 unique words, much larger than other literature
- Public release of data and code

**Weaknesses:**

- Potential overlap of test set with the training data of model since the test set is made from ACAV100M
- Audio Flamingo 2 hallucinates a lot and is an overall poor model for any real world task, using that to generate data is not good. The paper uses it not just for sound but also for music.
- Limited baselines - there are a lot more LALMs (open source and proprietary) models available in the literature
- Since a lot of pre-trained models are used to generate the benchmark a lot of biases can creep into the dataset

**Questions:**

- What is the logic or explanation behind the equations/coeffs in section 3.2?
- is there a correlation between DATE and human task preference?
- how does DATE correlate qualitatively with the model's performance on a captioning/QA task
- why no proprietary models?

---

> ### Author Response · Authors · 2025-11-29
> **Point-by-point response to Reviewer iKJk -- Part1**
>
> **Response**: We thank the reviewer for the valuable comments.
>
> > Reviewer iKJk (Weaknesses 1): Potential overlap of test set with the training data of model since the test set is made from ACAV100M.
>
> We agree that overlap with training data is an important concern, and MECAT was **explicitly designed to avoid reusing the dominant supervised caption sources.** **As stated in Section 3.1 of the paper**: “To **ensure data source novelty**, MECAT is constructed from a carefully selected subset of ACAV100M. This approach contrasts with benchmarks, such as AudioCaps, Clotho, and WavCaps-QA, which predominantly draw from a limited pool of sources such as AudioSet and Clotho **(see line 162-180 for Table 1 of the revised paper)**. The dataset comprises approximately 20,000 Creative Commons-licensed audio clips, each with a maximum duration of 10 seconds.”
>
> Most existing audio caption datasets are built from AudioSet segments that are widely reused for supervised training. In contrast, ACAV100M is a large-scale unsupervised corpus that, to the best of our knowledge, is **rarely used as supervised training data** for captioning/QA in current time. We first curate a ~20k CC-licensed subset from ACAV100M, remove near-duplicates, and then **re-annotate** all clips via our multi-expert + CoT pipeline; **no original ACAV100M titles/tags are copied** into MECAT. Thus, even if some raw waveforms were seen during pretraining, the fine-grained, multi-perspective captions and QA pairs in MECAT are **new from the label perspective**. To further support auditing of the potential overlap, we also release clip IDs/hashes.
>
> > Reviewer iKJk (Weaknesses 2): Audio Flamingo 2 hallucinates a lot and is an overall poor model for any real world task, using that to generate data is not good. The paper uses it not just for sound but also for music.
>
> > Reviewer iKJk (Weaknesses 4): Since a lot of pre-trained models are used to generate the benchmark a lot of biases can creep into the dataset
>
> Please refer to the Response to the **shared concern (ii) Data quality & potential bias**. Briefly, Audio Flamingo 2 is only one of several expert signals and is never treated as ground truth; conflicts with other experts are explicitly detected and filtered by our CoT-based synthesis and quality-control pipeline. We also conducted a human preference study, which confirms that MECAT references are strongly preferred over generic/incorrect alternatives and are on par with human-written captions.
>
> > Reviewer iKJk (Weaknesses 3): Limited baselines—there are a lot more LALMs (open source and proprietary).
>
> Please refer to the Response to the **shared concern (i) Insufficient baselines**. Briefly, we expanded the evaluation from 6 to 15 models on captioning and from 4 to 13 models on QA, including proprietary systems (Gemini-2.5 Pro/Flash), many open-weight LALMs (Qwen2.5-Omni 3B/7B, Qwen3-Omni, Audio Flamingo 3, Kimi-Audio-7B, Baichuan-Audio/Omni, Phi-4-Multimodal, step-audio-2-mini, Mimo-Audio), and caption-only baselines (EnClap, Pengi).

---

> > ### Author Response · Authors · 2025-11-29
> > **Point-by-point response to Reviewer iKJk -- Part2**
> >
> > > Reviewer iKJk (Question 1): What is the logic or explanation behind the equations/coeffs in Section 3.2?
> >
> > Please see Response to the **shared concern (v) Task-level weighting**. Briefly, all subtask scores are reported before any weighting, and the aggregated score uses fixed coefficients chosen to reflect task salience and data distribution.
> >
> > > Reviewer iKJk (Question 2): Is there a correlation between DATE and human task preference?
> >
> > Please refer to the Response to the **shared concern (iii) DATE design and alignment with human judgments**. Briefly, on the same 150 A/B caption pairs described earlier, human-preferred captions obtain a much higher DATE than non-preferred ones (mean 90.9 vs. 49.3), indicating a strong correlation between DATE and human preferences.
> >
> > > Reviewer iKJk (Question 3): How does DATE correlate qualitatively with the model’s performance on a captioning/QA task?
> >
> > Please refer to the Response to the **shared concern (iii) DATE design and alignment with human judgments**. Briefly, DATE combines a tf–idf–weighted semantic similarity term (how well a caption matches its own clip’s references) with a cross-sample discriminability term (whether it is reused across many different clips) to evaluted the open-ended caption / QA quality.
> >
> > Qualitatively, on captioning, models that produce clip-specific, detail-rich descriptions (e.g., about instrument timbre, speech intent, acoustic artifacts) obtain higher DATE, while models that recycle generic templates such as “a person is speaking” or “music is playing” are penalized by the discriminability term, even if they stay roughly on topic. On QA, the same pattern holds: DATE is higher when answers are tightly grounded in the audio and lower when they are generic or template-like. As a result, when a model improves both factual precision and clip-specific distinctiveness across MECAT, its DATE scores increase in a way that matches qualitative differences in performance.
> >
> > > Reviewer iKJk (Question 4): Why no proprietary models?
> >
> > Please refer to the Response to the **shared concern (i) Insufficient baselines**. Briefly, we have now included proprietary models Gemini-2.5 Flash and Gemini-2.5 Pro in both captioning and QA evaluations.

---

### Author Response · Authors · 2025-11-29
**General Response**

We are grateful to the AC for handling our submission under the current circumstances, and we thank all reviewers for their feedback, which has helped sharpen the work.

We understand that, due to the recent incident, reviewers’ scores and comments can no longer be updated. After submitting our rebuttal, we revised the paper and, to the best of our understanding, have addressed all of the concerns raised by the reviewers. Since the reviewers are no longer able to reflect these changes in their ratings or add follow-up comments, we summarize the current status for the AC.

Our work targets the evaluation gap highlighted in Section 1 of the revised paper: **existing audio understanding benchmarks, constrained by coarse annotations and non-discriminative metrics, often give similar scores to vague, generic responses and to detailed, clip-specific ones. On the data side, MECAT introduces fine-grained, multi-expert, multi-perspective annotations, and on the metric side, DATE explicitly rewards both semantic accuracy and cross-sample distinctiveness, so that benchmarks can distinguish generic outputs from highly detailed, clip-specific ones.** Reviewers also acknowledged our core contributions, including (i) the discriminative DATE metric that couples semantic similarity with cross-sample distinctiveness, (ii) a clear, reproducible multi-expert pipeline with explicit task/metric specifications, and (iii) MECAT’s utility as a fine-grained, multi-domain benchmark that surfaces model strengths and weaknesses.

Below we first summarize the main changes made to the paper in response to the reviews, and then indicate how they resolve the five shared concerns and the individual point-to-point questions. All modifications and additions to the manuscript are **highlighted in blue** text within the paper for easy identification.

**Main changes to the paper**

- **Expanded the original insufficient baselines and in-depth analysis.** We increased the evaluated model set to 15 systems (including Gemini 2.5 Pro/Flash, Qwen-Omni, Phi-4-Multimodal, etc.) and added slice-wise analysis by domain, Pure vs. Mixed, Short vs. Long captions, and QA sub-categories (Section 6.3.1, Tables 2-3).

- **Data quality and bias analysis.** We added human A/B preference studies, detailed pipeline descriptions, and analyses of domain coverage and potential biases, showing that MECAT achieves high preference rates vs. generic/incorrect baselines and parity with human references (Sections 5.1–5.2 and Appendix E).

- **Further explaination of DATE design and alignment with human judgments.** We clarified the embedding-level TF–IDF over contextual Sentence-BERT embeddings, motivated the corpus-relative discriminability term, and showed strong alignment with human preferences, while explicitly discussing remaining limitations of embedding-based metrics (Section 5.2 and Appendix E).

- **Scope and positioning of MECAT** We clarified the 10-second clip design, its suitability for fine-grained captioning/QA, and how MECAT complements existing large-scale benchmarks such as MMAU/MMAR (speech bias, silence hallucination, lack of acoustic-property reasoning; Sections 3–6).

- **Task-level weighting explanation.** We added a concise justification of the captioning and QA weighting scheme, linking coefficients to task importance and domain prevalence while keeping weights fixed across domains for comparability (Section 3.2).

In the following, we first address the **five shared concerns**: (i) Insufficient baselines, (ii) Data quality and potential bias, (iii) DATE design and alignment with human judgments, (iv) scope and positioning of MECAT (including clip length and relation to MMAU/MMAR), and (v) task-level weights; we then provide detailed point-to-point responses to each reviewer.

---

> ### Author Response · Authors · 2025-11-29
> **Shared concern: (i) Insufficient baselines -- Part1**
>
> > Reviewer iKJk (Weaknesses 3): Limited baselines - there are a lot more LALMs (open source and proprietary) models available in the literature
>
> > Reviewer iKJk (Question 4): why no proprietary models?
>
> > Reviewer jZBM (Weaknesses 3): For the experimental design part, only very few baselines are evaluated. This is not acceptable for a benchmark-focused paper -- which should check all existing open models as a bottom line (see how many baselines MMAR reported).
>
> > Reviewer sMGA (Weaknesses 2): The comprehensive of models evaluated is not great. Many more models have been released, including Audio Flamingo 3, etc.
>
> **Response**: We thank the reviewer for these valuable comments. We have expanded the evaluation **from 6 to 15 models on captioning** (2 classic caption-only models + 13 LALMs) and **from 4 to 13 models on QA**, spanning both **proprietary and open-weight families**. Proprietary coverage includes Gemini-2.5-Pro/Flash; open-weight families include Qwen (2.5-Omni-3B/7B, Qwen3-Omni), Audio Flamingo 3, Phi-4-Multimodal, Baichuan (Audio/Omni), Mimo-Audio-Instruct, step-audio-2-mini, Kimi-Audio-7B-Instruct, plus classic caption-only baselines Pengi and EnClap on the MECAT-Caption task. We hope this expanded coverage will address the baseline concern. These results are now included in the revised manuscript **(See Line 468-502 for the Table 2 and Table 3 in the Revised paper)**.

---

> ### Author Response · Authors · 2025-11-29
> **Shared concern: (i) Insufficient baselines -- Part2**
>
> **Table: Model Performance (DATE \%) on MECAT-Caption.**
> *Note: **Bold** indicates the best performance, and $\underline{underline}$ indicates the second best. $^\dagger$ indicates that the model or its previous version was explicitly used in the data construction process.*
>
> | Type         | Model                     | Systemic (Long)    | Systemic (Short)   | Speech (Pure)      | Speech (Mixed)     | Music (Pure)       | Music (Mixed)      | Sound (Pure)       | Sound (Mixed)      | Env                | Score (Cap)        |
> | :---         | :---                      | :---:              | :---:              | :---:              | :---:              | :---:              | :---:              | :---:              | :---:              | :---:              | :---:              |
> | Caption-Only | Pengi                     | 43.5               | 46.8               | 27.2               | 29.5               | 29.3               | 13.1               | 42.8               | 14.6               | 7.1                | 29.4               |
> |              | EnClap                    | 48.6               | 53.1               | 30.2               | 31.8               | 17.9               | 15.9               | 48.8               | 15.2               | 6.8                | 31.9               |
> | LALM         | Phi-4-Multimodal-Instruct | 42.4               | 44.0               | 26.9               | 31.3               | 14.9               | 24.0               | 28.5               | 18.1               | 13.1               | 30.0               |
> |              | Kimi-Audio-7B-Instruct    | 49.5               | 54.2               | 30.0               | 31.3               | 27.7               | 16.9               | 43.1               | 16.2               | 7.0                | 32.8               |
> |              | Baichuan-Audio-Instruct   | 42.6               | 36.5               | 46.0               | 40.4               | 21.3               | 20.7               | 44.8               | 17.7               | 15.1               | 33.7               |
> |              | Audio Flamingo 2{\dagger}         | 48.6               | 49.7               | 30.5               | 34.3               | 28.8               | 25.6               | 41.2               | 18.5               | 17.5               | 35.3               |
> |              | Baichuan-Omni             | 47.0               | 50.9               | 43.5               | 41.7               | 35.2               | 13.7               | 34.3               | 19.7               | 11.3               | 35.6               |
> |              | Mimo-Audio-Instruct       | 56.5               | 56.9               | 45.8               | 44.9               | 35.8               | 19.4               | 46.8               | 21.0               | 9.8                | 40.1               |
> |              | Audio Flamingo 3{\dagger}         | 52.5               | 51.5               | 49.3               | $\underline{48.8}$ | 40.4               | 24.8               | 50.6               | 21.9               | 11.5               | 40.4               |
> |              | Qwen3-Omni                | 47.9               | 43.7               | 50.2               | 48.7               | 51.2               | 26.8               | 49.0               | 19.5               | 18.2               | 40.4               |
> |              | Step-audio-2-mini         | 55.6               | 58.7               | 44.2               | 43.6               | 35.3               | 32.0               | 42.8               | 18.9               | 16.1               | 41.5               |
> |              | Qwen2.5-Omni 3B           | 56.4               | 55.2               | 42.5               | 41.3               | 46.6               | 29.7               | $\underline{52.9}$ | 23.9               | 19.4               | 42.5               |
> |              | Qwen2.5-Omni 7B           | 61.1               | 56.5               | 39.9               | 40.9               | 32.1               | 30.9               | 50.7               | 23.8               | 17.9               | 42.6               |
> |              | Gemini-2.5-Flash          | **65.6** | **63.9** | **57.5** | **57.5** | $\underline{52.9}$ | **41.0** | 52.2               | $\underline{28.3}$ | $\underline{22.1}$ | **51.6** |
> |              | Gemini-2.5-Pro            | $\underline{62.3}$ | $\underline{62.4}$ | $\underline{56.6}$ | **57.5** | **53.6** | $\underline{38.7}$ | **53.4** | **29.9** | **24.0** | $\underline{50.6}$ |

---

> ### Author Response · Authors · 2025-11-29
> **Shared concern: (i) Insufficient baselines -- Part3**
>
> **Table: Model Performance (DATE %) on MECAT-QA.**
> *Note: **Bold** indicates the best performance, and $\underline{underline}$ indicates the second best. $^\dagger$ indicates that the model or its previous version was explicitly used in the data construction process.*
>
> | Model                     | Direct Perception | Sound Characteristics | Quality Assessment | Environment Reasoning | Inference & Judgment | Application Context | Score (QA)         |
> | :---                      | :---:             | :---:                 | :---:              | :---:                 | :---:                | :---:               | :---:              |
> | Kimi-Audio-7B-Instruct    | 45.6              | 39.2                  | 18.7               | 34.6                  | 48.9                 | 41.2                | 38.0               |
> | Baichuan-Audio-Instruct   | 40.7              | 45.2                  | 31.0               | 35.1                  | 49.0                 | 46.9                | 41.3               |
> | Audio Flamingo 2$^\dagger$| 45.1              | 46.3                  | 34.9               | 37.5                  | 44.0                 | 42.4                | 41.7               |
> | Baichuan-Omni             | 43.6              | 44.7                  | 33.7               | 39.9                  | 49.3                 | 49.1                | 43.4               |
> | Phi-4-Multimodal-Instruct | 48.4              | 46.3                  | 34.7               | 40.2                  | 49.3                 | 48.7                | 44.6               |
> | Mimo-Audio-Instruct       | $\underline{59.3}$| 49.3                  | 24.9               | 39.1                  | 52.7                 | 46.2                | 45.2               |
> | Step-Audio-2-mini         | 57.7              | 54.3                  | 37.2               | 39.2                  | 48.9                 | 48.0                | 47.6               |
> | Audio Flamingo 3$^\dagger$| 53.8              | 50.2                  | 36.0               | 43.0                  | 54.5                 | 49.6                | 47.8               |
> | Qwen2.5-Omni 3B           | 55.7              | 53.2                  | 38.6               | 41.1                  | 51.8                 | 50.8                | 48.5               |
> | Qwen2.5-Omni 7B           | 57.8              | 52.9                  | $\underline{39.1}$ | 44.0                  | 53.2                 | 50.8                | 49.6               |
> | Qwen3-Omni                | **61.7** | $\underline{54.6}$    | **39.3** | 45.0                  | 56.9                 | 56.1                | **52.3** |
> | Gemini-2.5-Flash          | 56.3              | **55.3** | 37.7               | $\underline{46.8}$    | **58.6** | **58.0** | $\underline{52.1}$ |
> | Gemini-2.5-Pro            | 55.5              | 54.4                  | 37.7               | **47.6** | $\underline{57.3}$   | $\underline{56.6}$  | 51.5               |

---

> ### Author Response · Authors · 2025-11-29
> **Shared concern: (ii) Data quality & potential bias -- Part1**
>
> > Reviewer iKJk (Weaknesses 2): Audio Flamingo 2 hallucinates a lot and is an overall poor model for any real world task, using that to generate data is not good. The paper uses it not just for sound but also for music.
>
> > Reviewer iKJk (Weaknesses 4): Since a lot of pre-trained models are used to generate the benchmark a lot of biases can creep into the dataset.
>
> > Reviewer 5m6Q (Weaknesses 1): The construction stack depends on ASR, diarization, emotion, and event taggers. Mistakes at these stages can leak into captions and QA despite later filtering. The paper notes filtering, yet residual noise is likely in complex audio. This critiques robustness, not the clarity of the written specification.
>
> > Reviewer jZBM (Weaknesses 1): For the data construction, since all steps are based on AI models and there is no human-in-the-loop, it is questionable whether the outputs are high-quality enough or have some hidden bias. Since none of the AI models are super reliable to my knowledge, I tend to believe such biases or inaccurate test samples exist. Therefore, this work should at least verify the quality of the testset via human inspection or tests.
>
> **Response**: We thank the reviewers for these valuable comments. Both caption and QA labels in MECAT are generated via a pipeline that integrates analysis from specialized expert models (ASR, diarization, event/music taggers, acoustic analyzers, etc.) with a **Chain-of-Thought LLM synthesis stage** that jointly reasons over these signals, explicitly $\underline{\rm detects\ conflicts}$, and either $\underline{\rm reconciles\ the\ evidence}$ or $\underline{\rm discards\ unsupported\  claims}$ to produce multi-perspective, fine-grained annotations. To limit automation bias and upstream errors, we perform multi-source cross-checking with confidence/consistency checks and hallucination removal.
>
> Because these expert models are architecturally and training-wise diverse, their errors are largely uncorrelated; compared with a single-model pipeline, the probability that they all hallucinate the same content on a given clip is substantially lower. In a quantitative audit focusing on **conflicts** between Audio Flamingo 2 and the other experts, **fewer than 0.5\%** of the released clips were flagged as still having any potential unresolved conflict after the full pipeline. Manual inspection of a subset of these flagged cases confirmed that CoT had already removed most unsupported Audio Flamingo 2 details and retained only the consistent core content, as in the following example from the MECAT-Caption set:
>
> One clip exhibited clear conflicts across models: Audio Flamingo 2 described `a vehicle engine runs,` while ASR and diarization contained only instructional speech, and CED-Base predicted `Speech` plus a single `Insect` tag that no other source supported.
>
> CoT conflict-resolution:
>
>
> There's a conflict between Audio Flamingo's `vehicle engine` and the lack of such tags in CED audio tags. Also, the `Insect` label in CED isn't supported elsewhere ... Synthesis Guidelines require prioritizing consistent data. The conflicting engine/insect sounds and the metadata can't be used. ... Overall, confidence is High because the primary elements are consistent.
>
>
> Resulting captions:
>
> Short: `Male instructor explaining technical procedures with background interference.`
>
> Long: `A male speaker with American accent provides detailed technical instructions about proper equipment formation, with consistent background interference affecting audio clarity.`

---

> ### Author Response · Authors · 2025-11-29
> **Shared concern: (ii) Data quality & potential bias -- Part2**
>
> In addition, we validate label trust worthiness via **a human preference study**. We conducted an A/B test on 150 caption pairs spanning all domains, where listeners selected the better caption under the criterion `accuracy first, then level of detail.` MECAT references were strongly preferred over both Safe captions (generic, vague descriptions, e.g., `A man is speaking` reused for all speech-only audio) and Wrong captions (factually incorrect references randomly selected from other samples), and were **statistically on par with Human references** under the same instruction. We will include this human preference study in the revised manuscript. **(See line 390-394 for Section 6.1 and Line 1302-1325 for Appendix F.1 of the revise d paper)**
>
> **Table: Human Preference A/B Validation (150 pairs)**
> | Opponent type (B) |   n | Reference (A) win rate | 95% CI (Wilson) |
> | ----------------- | --: | ---------------------: | --------------: |
> | Overall           | 150 |                  82.7% |  [75.8%, 87.9%] |
> | Safe              |  52 |                  94.2% |  [84.4%, 98.0%] |
> | Wrong             |  47 |                  97.9% |  [88.9%, 99.6%] |
> | Human             |  51 |                  56.9% |  [43.3%, 69.5%] |

---

> ### Author Response · Authors · 2025-11-29
> **Shared concern: (iii) DATE design and alignment with human judgments -- Part1**
>
> > Reviewer iKJk (Question 2): Is there a correlation between DATE and human task preference?
>
> > Reviewer iKJk (Question 3): How does DATE correlate qualitatively with the model’s performance on a captioning/QA task?
>
> > Reviewer n5Pw (Question 3): What will DATE represent, or mean, on a dataset level?
>
> > Reviewer 5m6Q (Question 2): Please explain why DATE combines a tf–idf weighted semantic similarity term with a cross-sample discriminability term, and why you fuse them with a harmonic mean. Which failure modes in audio captioning does this design aim to fix, and why prefer these components over alternatives such as pure embedding similarity or learned weighting.
>
> > Reviewer jZBM (Weakness 2): For the metric design part, while I agree that current captioning-based metrics (CIDEr, SPICE, FENSE, etc) are not good enough, I do not think the proposed DATE addresses the issues. ... However, we expect the metric to be fixed for a given sample, regardless what the other samples are.
>
> **Response**: We thank the reviewers for these valuable comments. DATE is designed to **address two concrete failure modes** we observed: First,  $\underline{\rm generic-but-vague\ captions}$ that stay roughly on topic but are recycled across many clips, and second, $\underline{\rm over-specific\ but incorrect\ captions}$. For each clip, DATE is defined as the harmonic mean of the embedding-based tf–idf–weighted semantic similarity to that clip’s reference captions and the cross-sample discriminability term, so a caption must be both semantically faithful and clip-specific to obtain a high score. **As illustrated in Figures 2 and 4 **(See line 201-215 for Figure 2 and Line 358-377 for Figure 4 in the revised paper)**, MECAT spans a wide range of speech, music, and environmental sound domains; in such a diverse corpus, a system that produces very similar captions for many clips will receive low discriminability scores and thus low DATE, even if those captions are loosely on topic **(See line 401-419 for Figure 5 of the revised paper)**. At the dataset level, DATE is simply the average of per-sample DATE over all clips, so higher dataset-level DATE indicates that a system, on average, **produces captions that are accurate with respect to MECAT references and non-generic across the corpus**.
>
> We apologize for not explaining the TF–IDF computation clearly in the original submission. The revised paper now adds **Appendix E**, which details the *embedding-level* TF–IDF scheme used in DATE and clarifies that it operates on **contextual Sentence-BERT token embeddings**, not static type-level vectors. Sentence-BERT produces contextualized token embeddings whose representations explicitly depend on the surrounding sentence. DATE then applies embedding-level TF–IDF weights $w(t, T)$ to these contextual embeddings and constructs the sentence representation as described in **Section 6.2 (see line 330-340)**. This makes DATE’s similarity term more expressive than a pure bag-of-words count over static embeddings: words that are central in a given context receive larger weights, and semantically related variants (e.g., “dog” / “canine”) can be treated consistently in embedding space. To prove DATE is not BoW, we compared a Gemini-2.5-Pro caption (Long) with its shuffled version. **The dramatic drop in DATE ($62.3 \rightarrow 35.0$) confirms sensitivity to word order, decisively proving DATE is not BoW**.
>
> However, we agree with Reviewer jZBM that embedding-similarity-based term inevitably smooths over some fine-grained semantic distinctions and does not explicitly encode the temporal structure of the underlying audio. This limitation is exactly what we observe in **Section 6.2 and Figure 5 (lines 397–420)**: the similarity term alone often fails to sharply separate good and bad captions. This observation motivates the design of DATE’s discriminability component and the use of the harmonic mean: the harmonic mean penalizes systems that are high on similarity but low on discriminability (generic templates), or vice versa (idiosyncratic but incorrect captions), and forces high DATE only when both terms are simultaneously strong (addressing Reviewer 5m6Q’s question about design choices).
>
> While we understand the intuition that a metric should be fixed for a given sample, DATE is intentionally corpus-relative. Our goal is not only to measure how well a caption matches its own references, but also to detect the failure mode where a model reuses generic templates across many clips. This requires comparing each predicted caption against other clips in the benchmark. For a given MECAT release and a given model (i.e., its outputs on all clips), the corpus used to compute the similarity and discriminability terms is fixed, so both terms and the resulting per-sample and dataset-level DATE scores are deterministic. We view this corpus-relative design as essential for measuring distinctiveness, rather than as a limitation.

---

> > ### Author Response · Authors · 2025-11-29
> > **Shared concern: (iii) DATE design and alignment with human judgments -- Part2**
> >
> > To assess **alignment between DATE and human judgments**, we used the same 150 A/B caption pairs as in (ii), spanning all MECAT domains and opponent types (Safe, Wrong, Human as detailed in (ii)). Listeners were asked to choose the better caption under the rule `accuracy first, then level of detail.` Human-preferred captions received much higher DATE scores than non-preferred ones (90.9 vs 49.3 on average), indicating that DATE is **strongly correlated with human preferences** under this instruction **(see line 440-434 for Section 6.2 and line 1327-1345 for Appendix F.2 of the revised paper)**.
> >
> > **Table: Alignment between DATE and human preferences (150 A/B caption pairs)**
> >
> > | Quantity                                  | Value |
> > |-------------------------------------------|-------|
> > | Mean DATE of human-preferred captions     | 90.9  |
> > | Mean DATE of non-preferred captions       | 49.3  |

---

> > > ### Author Response · Authors · 2025-11-29
> > > **Shared concern: (iv) Scope and positioning of MECAT (10-second clips and relation to MMAU/MMAR)**
> > >
> > > > Reviewer n5Pw (Weakness 3): For audio understanding tasks, one of the problems in current audio LMs is that the length of audio they can process (before a significant performance drop) is very short, usually less than 30 seconds. Since this work focuses on understanding more nuanced, more descriptive content for each audio clip, I think the 10-second audio length in MECAT is very limited. This is one major drawback from the perspective of the contribution of the work.
> > >
> > > > Reviewer sMGA (Weakness 1): The maximum duration is only 10 seconds. Short audio snippets can only be as complex.
> > >
> > > > Reviewer sMGA (Weakness 3): The tasks for coverage are very foundational and have been explored earlier. So I find a slight lack of novelty here.
> > >
> > > > Reviewer jZBM (Weakness 1): Furthermore, there is no solid evidence the proposed benchmark is siginificantly more useful than MMAU/MMAR, despite it being open-ended.
> > >
> > > > Reviewer jZBM (Weakness 3): There is not much analysis on different audio language models given the scores. What do the scores imply besides simply ranking the models? Do the scores reveal any potential bias of existing models? Do the scores show noticeable hints that prior benchmarks (MMAU, MMAR) did not show? Are there PEFT or training-free methods to improve baseline models on the benchmark? These are some examples that can show the usefulness of the proposed benchmark.
> > >
> > > **Response:** We thank the reviewers for these valuable comments. Our goal with MECAT is to **complement** existing benchmarks such as MMAU/MMAR by targeting a different failure mode: models that achieve reasonable multiple-choice accuracy without producing grounded, informative descriptions of the audio. MMAU/MMAR are primarily QA-style and often multiple-choice; in such settings, models can rely heavily on the textual question/option priors, making non-trivial accuracy possible even when the audio input is weakly informative (e.g., white noise or heavily degraded audio). MECAT instead uses **open-ended captioning and QA**, together with DATE, to require models to generate accurate and discriminative descriptions: simply guessing from priors or reusing generic templates will not score well because (i) captions are compared to rich, multi-perspective references and (ii) the cross-sample term explicitly penalizes captions that “fit” many clips equally well. This open-ended, discriminative setup is precisely where we see current audio LMs struggle in our experiments, even when they perform reasonably on prior multiple-choice benchmarks.
> > >
> > > The choice of 10-second clips is **intentional**. Many long-audio datasets provide only **coarse, global labels** for very long inputs. For example, in https://huggingface.co/datasets/nvidia/LongAudio/viewer/default/gigaspeech?row=1, a one-hour podcast may be annotated with a single summary such as `The podcast discusses investigations related to President Trump, political polarization, impeachment, and educational admissions controversies.` Such labels do not specify which parts of the audio support which claims, so a model can learn to reproduce high-level topics **without truly grounding fine-grained content** in the signal. MECAT takes the opposite design: each 10-second clip carries 18 multi-perspective captions and 5 open-ended QA pairs covering speech, music, sound events, and acoustic properties. This **dense, local supervision** forces models to capture nuanced, clip-level information, while the short duration keeps the setting compatible with current LALM context limits and makes clip-level discriminability (as required by DATE) well defined. Long-audio robustness remains important and is better covered by other existing benchmarks such as LongAudioBench; MECAT adds a **complementary axis** that stresses detailed, discriminative understanding at the local level. Our aim is that MECAT guides future model and pretraining design toward better disentangling and jointly modeling speech, music, and general sound, rather than optimizing various length coverage under sparse, global labels **(see line 189-190 for Section 3.1 and line 534-539 for Section 7 of the revised paper)**.

---

> ### Author Response · Authors · 2025-11-29
> **Shared concern: (v) Task-level weighting**
>
> > Reviewer iKJk (Question 1): What is the logic or explanation behind the equations/coeffs in Section 3.2?
>
> > Reviewer 5m6Q (Weakness 3): Caption weights are fixed, and the QC pipeline uses an empirical GLAP threshold. The formulas are clear, yet the motivation for these constants is brief, which can affect reproducibility across domains.
>
> > Reviewer 5m6Q (Question 1): What design principle determined the caption category and subcategory weights, and are these weights intended to reflect perceived task importance or observed reliability of references. Please clarify whether weights are constant across domains and why.
>
> **Response:**  We thank the reviewers for these valuable comments. All subtask scores (caption slices and QA categories) are reported before any weighting; the weighted score is only a single headline number for quick comparison.
>
> For captioning, the top-level weights 0.4 / 0.4 / 0.2 on Systemic, Content-Specific, and Environment reflect three complementary aspects of information: overall description, content detail, and contextual background, with Systemic and Content-Specific treated as primary and Environment as supporting. Within Systemic, long and short captions are weighted 0.8 / 0.2 because long captions carry more evaluative signal. Within Content-Specific, Speech / Music / Sound are weighted 0.6 / 0.3 / 0.1 to roughly follow their relative prevalence in ACAV100M. For each content type, pure and mixed domains are averaged equally to value robustness in both clean and composite scenes. For QA, the six subcategories (Direct Perception, Sound Characteristics, Quality Assessment, Environment Reasoning, Inference & Judgment, Application Context) are averaged uniformly (weight 1/6 each), since they are intended as equally important facets of audio understanding. All weights are fixed across domains to preserve comparability, and we verified that model rankings are qualitatively stable under reasonable perturbations of these coefficients. **(see line 232-235 for Section 3.2 of the revised paper)**

---

### Meta-Review · Area_Chair_V2hT · 2026-01-06

**Summary:**

Reviewers identified four main shared key concerns. The issue of insufficient baselines (reviewers iKJk, jZBM, sMGA) was addressed in the rebuttal through additional experiments. Concerns about data quality and bias from AI-generated evaluation data (reviewers iKJk, 5m6Q, jZBM) remain unresolved, despite mitigation strategies discussed in the rebuttal. Issues regarding the novelty and technical depth of the DATE metric and its alignment with human judgments (reviewers iKJk, n5Pw, 5m6Q, jZBM) were partially addressed, but limitations remain. Finally, concerns about the scope of the MECAT benchmark, particularly the use of fixed 10-second clips (reviewers sMGA, jZBM), were not fully addressed.

Hence AC's recommendation of this paper is Reject.

**Reviewer Concerns:**

Concern 1: Insufficient baselines
(Reviewers iKJk, jZBM, sMGA)

- This concern was addressed during the rebuttal, where the authors provided experimental results with additional baselines.

Concern 2: Data quality and potential bias from AI-generated data
(Reviewers iKJk, 5m6Q, jZBM)

- This concern was not fully addressed. While the use of AI-generated data may be reasonable for scaling training data, its suitability for evaluation benchmarks remains questionable. Although the rebuttal discussed mitigation strategies such as CoT-based filtering, concerns about evaluation validity and bias persist.

Concern 3: DATE metric design and alignment with human judgments
(Reviewers iKJk, n5Pw, 5m6Q, jZBM)

- This concern was partially addressed. The rebuttal clarified aspects of the DATE metric and its alignment with human judgments; however, questions regarding the metric’s novelty and technical contribution remain.

Concern 4: Scope and positioning of MECAT
(Reviewers sMGA, jZBM)

- This concern was not addressed. The restriction to fixed 10-second clips introduces potential bias.

**Reviewer Scores:**

Reviewer iKJk: Likely to increase their score from 2 to 4 following the rebuttal, though a score above 6 would still be unlikely.

Reviewer n5Pw: Unlikely to change their score; expected to remain at 4.

Reviewer 5m6Q: Unlikely to change their score; expected to remain at 6.

Reviewer sMGA: Unlikely to change their score; expected to remain at 4.

Reviewer jZBM: Unlikely to change their score; expected to remain at 2.

---

### Decision · Program_Chairs · 2026-01-26

Reject